# Evaluating the accuracy of a nutritional screening tool for patients with digestive system tumors: A hierarchical Bayesian latent class meta-analysis

**Menghao Yang[1]⊙, Na Xiao[1]⊙, Le Tang[1], Yang Zhang[1], Yuexiu Wen[2], Xiuqin Yang[1,2]***

**1** School of Nursing Guizhou Medical University, Guiyang, Guizhou, China, **2** Department of General Surgery, The Second Affiliated Hospital of Guizhou Medical University, Kaili City, Qiandongnan Prefecture, Guizhou Province, China

⊙ These authors contributed equally to this work.

* 3129863401@qq.com

## Abstract

### Background

Cancer, particularly tumors of the digestive system, presents a major global health challenge. The incidence and mortality rates of these cancers are increasing, and many patients face significant nutritional risks, which are often overlooked in clinical practice. This oversight can lead to serious health consequences, underscoring the need for effective nutritional assessment tools to improve clinical outcomes. Although several nutritional risk screening tools exist, their specific utility for patients with gastrointestinal tumors remains unclear. This study aimed to address this gap by systematically evaluating the performance of various nutritional screening tools in this patient population.

### Methods

A systematic search of six databases was conducted to identify studies that met predefined inclusion and exclusion criteria. Diagnostic test metrics such as sensitivity, specificity, and likelihood ratios (positive and negative) were estimated using a hierarchical summary receiver operating characteristic model. This approach was used to compare the accuracy of different nutritional screening scales.

### Results

A total of 33 eligible studies were included in this meta-analysis, assessing six nutritional screening tools: the Malnutrition Universal Screening Tool, Malnutrition Screening Tool, Nutritional Risk Screening 2002, Mini Nutritional Assessment-Short Form, Nutritional Risk Index, and Patient-Generated Subjective Global Assessment. Among these, the Patient-Generated Subjective Global Assessment demonstrated the highest performance, with a sensitivity of 0.911 (95% confidence interval: 0.866–0.942) and a specificity of 0.805 (95% confidence interval: 0.674–0.891), outperforming the other screening tools.

**Data Availability Statement:** All relevant data for this study are publicly available from the Dryad

repository (https://doi.org/10.5061/dryad.4mw6m90m8).

**Funding:** This study received funding from the Science and Technology Fund Project of the Health and Wellness Commission of Guizhou Province, China, under Project No. [gzwkj2024-266]. The funders had no role in study design, data collection and analysis, decision to publish, or preparation of the manuscript.

**Competing interests:** The authors assert that there are no conflicts of interest in the conduct of this study.

## Conclusions

This study confirms the effectiveness of the Patient-Generated Subjective Global Assessment in identifying malnutrition risk among patients with digestive system tumors. However, as this research focused on a Chinese population, future studies should encompass a broader geographic scope and work toward standardized assessment criteria to enhance the global validation and refinement of nutritional screening tools.

## Introduction

Cancer is one of the major threats to global health, and governments have made cancer control an important element of their public health strategies [1]. According to GLOBOCAN cancer data, in 2020, there will be approximately 19.3 million new cancer cases and 10 million cancer deaths worldwide. By 2040, the number of new cancer cases worldwide is expected to increase to 28.4 million, a 47% increase from 2020 [2,3]. Among different types of cancers, tumors affecting the digestive system encompass esophageal, stomach, colorectal, pancreatic, and liver cancers. These tumors are highly malignant and are associated with extremely high morbidity and mortality rates, accounting for more than half of the global tumor-related morbidity and mortality. According to the statistical analysis report published by the Journal of Cancer for Clinicians in 2021, three cancers of the digestive system: colorectal, liver, and gastric cancers, ranked second, third, and fourth, respectively, in terms of major global incidence and mortality rates. Yearly, there is a rising occurrence and death rate associated with tumors of the digestive system, posing a significant risk to the population's life and well-being. They constitute a crucial factor resulting in the death of patients and pose a great challenge to public health [4,5].

Nutritional risk is a common but frequently overlooked problem in patients with tumors. Studies have shown that approximately 40%–80% of patients with malignant tumors are at nutritional risk, with approximately 20% of this population dying directly owing to malnutrition [6]. The European Society for Parenteral Nutrition considers nutritional risks to be existing factors that may adversely affect a patient's clinical outcome, resulting in an increased risk of infections and complications and prolonged hospitalization [7]. Patients with tumors are prone to malnutrition due to the disease and the side effects of its treatment, which manifests as loss of appetite, nausea, vomiting, fatigue, pain, and other symptoms [8]. The incidence of malnutrition is related to the type and location of the tumor and is particularly highest in patients with malignant tumors of the digestive system. Relevant studies have shown that the incidence of malnutrition in patients with malignant tumors of the digestive system accounts for 60% of the incidence of malnutrition in all patients with malignant tumors [9]. As the disease progresses, prolonged failure to improve the nutritional status of patients with tumors can lead to further deterioration of the disease [10]. It is now widely accepted that prevention of cancer cachexia is more effective than treatment and that the best management is early detection of malnutrition through nutritional risk screening and timely intervention through nutritional therapy [11].

Currently, no recognized "gold standard" exists for nutritional risk screening methods. Existing nutritional risk screening methods are widely used in clinical practice. However, they vary in their focus, sensitivity, specificity, and operability [12,13]. Traditional nutritional screening assessment methods, such as anthropometric indices, are straightforward to use but may lack reliability and efficiency in quickly measuring patients' nutritional status [14]. Several comprehensive nutritional risk screening scales, such as Nutritional Risk Screening 2002 (NRS-2002), Malnutrition Universal Screening Tool (MUST), Mini Nutritional Assessment-Short Form (MNA-SF), Nutritional Risk Index (NRI), and Malnutrition Screening Tool

(MST), are presently employed in clinical settings. However, most of these nutritional risk screening scales are universal screening tools that lack customization for specific diseases [15]. For example, the NRS 2002 is a commonly used nutritional risk screening tool, but it does not consider individual differences between patients with specific types of cancer. The NRS 2002 fails to differentiate between patients with gastric cancer and those with colon cancer, who may have different nutritional challenges and needs. [16]. Similarly, the MNA-SF is primarily used to assess the nutritional status of older adults and may not accurately reflect nutritional risk in younger patients, whose physiological status and nutritional needs differ significantly from those of older adults [17]. Additionally, MUST provides a rapid assessment of an individual's nutritional status but may overlook other nutrition-related indicators that are equally important for patients with cancer [18]. Therefore, these existing nutritional risk screening scales, while providing some clinical reference, lack specificity for a particular disease or patient group and do not enable a comprehensive assessment of a patient's nutritional status.

Inclusive assessment tools are essential for evaluating the nutritional status of patients with digestive system tumors, as these malignancies pose unique challenges that complicate nutritional management. In the absence of a universally accepted gold standard for nutritional assessment, applying rigorous statistical methods is crucial for improving diagnostic accuracy. The hierarchical summary receiver operating characteristic (HSROC) model is particularly well-suited for this purpose, as it addresses variations in reference standards and accounts for differences in diagnostic accuracy across studies. This model enables researchers to account for variability in test results, making it a valuable asset in our analysis [19–22]. To address these challenges, we performed a hierarchical Bayesian latent class meta-analysis, synthesizing existing literature to systematically compare the efficacy of various nutritional screening tools. This comprehensive analysis specifically examines the validity and applicability of these tools for patients with digestive system tumors. However, it is important to recognize that our findings may be limited by the regional focus of the included studies, predominantly from China, which could affect the generalizability of our conclusions. Our ultimate objective is to improve clinical approaches to nutritional assessment, thereby enhancing patient outcomes and quality of life. By offering a detailed comparison of available screening tools, we aim to inform clinical practice and guide future research in this vital area of patient care

## Materials and methods

### Search strategy and selection criteria

PubMed, Embase, Cochrane Database of Controlled Clinical Trials, China Knowledge, and Wanfang and Wikipedia databases were searched to collect diagnostic tests on the accuracy of nutritional screening questionnaires in patients with digestive system tumors. The searches spanned from the inception of the databases to February 29, 2024. The search methodology included a combination of subject terms and free words adapted to the search characteristics of each database. Search terms included "Digestive System Neoplasms [MESH]," "Cancers of the Digestive System," "Neoplasms of the Digestive System," "Biliary Tract Neoplasms," "Gastrointestinal Neoplasms," "Liver Neoplasms," "Liver cancer," "Pancreatic Neoplasms," "Pancreatic cancer," "Esophageal Neoplasms," "Carcinoma of the esophagus," "Intestinal Neoplasms," "Colorectal cancer," "Stomach Neoplasms," "Gastric cancer," "Nutrition Assessment," "Nutritional Assessments," "Nutritional Risk Screening 2002," "Patient-generated Subjective Global Assessment," Universal Malnutrition Screening Tool," "Malnutrition Screening Tool," "Mini Nutritional Assessment," "Mini Nutritional Assessment-Short Form," "PG-SGA," "MNA," "NRS-2002," "SGA," "MUST," "MST," and "MNA-SF." Concurrently, the citations of the included papers were searched to gather additional pertinent information.

Following the elimination of duplicate publications, we established logical inclusion and exclusion criteria to assess the remaining studies. Two researchers autonomously evaluated the incorporation and exclusion of each study based on the established criteria. Any disagreements were settled during a consensus meeting. If a unanimous agreement could not be achieved, a third researcher was consulted to settle the matter. The inclusion criteria were as follows: (1) The participants in the study were individuals diagnosed with tumors in the digestive system, such as colorectal cancer, gastric cancer, oesophageal cancer, biliary tract tumors, pancreatic cancer, and hepatocellular carcinoma; it did not matter if they had undergone surgery, chemotherapy, or radiotherapy, or if they had been hospitalized or not. (2) The study was a diagnostic study. (3) The reference standards used in the study included those from the European Society of Clinical Nutrition and Metabolism and the Global Malnutrition Leadership Initiative, as well as subjective global assessment, patient-generated subjective global assessment (PG-SGA), albumin, prealbumin, and objective assessments by other professionals. (4) Nutritional screening tools used in the study included "PG-SGA," "MNA," "NRS-2002," "SGA," "MUST," "MST," and "MNA-SF," among others. (5) The study included original research that could be obtained directly or indirectly, and the data collected included true positive (TP), false positive (FP), false negative (FN), true negative (TN), and other data. Exclusion criteria: (1) type of study does not meet the definition of a diagnostic study, e.g., case report, review article, non-clinical study, etc.; (2) published in non-English or non-Chinese language: the study has not been published in English or Chinese; (3) full text unavailable: the full text of the study is unavailable for detailed assessment despite reasonable attempts to do so; (4) incomplete or unextractable data, information about the results or methods of the Missing information

## Data extraction

The data extraction process was performed by two researchers who were separately responsible for the data extraction of the included studies. They also conducted a thorough assessment of the quality and similarity of the data collected. Any disagreements that arose were resolved either in consensus meetings or, if necessary, referred to a third party for resolution. The data extractions comprised the following three main components: (1) details pertaining to the original study, including the primary author, publication year, and the author's country; (2) characteristics of the study, such as sample size, sex distribution, age, disease diagnosis, nutritional screening methods, and standards of reference; and (3) study outcomes, which included TP, FP, FN, and TN.

In analyzing multiple publications of a study, we ensured that data were included only once to prevent any overlap. For the purposes of the survey, we chose datasets with large sample sizes or recently published research. Additionally, we included all instances of the same evaluation instrument cited several times within a single study in our analysis. This approach ensured that we captured all relevant information while avoiding any redundancy.

## Literature quality assessment and publication bias

Literature quality was assessed using the Quality Assessment of Diagnostic Accuracy Studies (QUADAS) checklist, and methodological quality maps and methodological quality map summaries were generated using Review Manager 5.3 software. Furthermore, the presence of publication bias in the included literature was determined by the Deek test of Stata 17.0.

## Data analysis and statistical methods

A series of TP, FP, TN, and FN data were calculated by transforming the results from the included literature into dichotomous variables (risk of being malnourished and risk of not

being malnourished). Sensitivity, specificity, positive and negative likelihood ratios, and corresponding 95% confidence intervals (95% CIs) were jointly estimated using the HSROC model. The variability of reference standards between studies is one of the factors due to the lack of a gold standard for evaluating malnutrition. Evidence suggests that ignoring the imperfect nature of reference standards in meta-analyses of screening accuracy may lead to significant bias in the combined estimates [23]. The traditional Moses-Lenberg approach fails to model appropriately between study variability to account for uncertainty in the estimates [24]. To address this problem, Rutter and Gatsonis [19] proposed the HSROC model as a basic framework for meta-analyses of diagnostic tests, which can be fitted by a fully Bayesian approach. Macaskill highlighted [25] that applying an empirical Bayesian approach to fit the HSROC model yields similar results to the full Bayesian approach because of its simplicity and ease of implementation, which can be used as an alternative method. Moreover, Chu and Cole [26] have suggested a logarithmic conversion of metrics, such as sensitivity and specificity, to align with the model using a simple linear mixed model approach, which is comparable to Macaskill's empirical Bayesian method. Building upon Chu and Cole's [26] basic linear mixed model approach, Harbord et al. developed the "metandi" package to fit the HRSOC model with a two-level mixed logistic regression model using Stata software. We used the HSROC model to examine the sensitivity, specificity, diagnostic ratio, likelihood ratio (LR)+ and LR- of the combined diagnostic tests. Using this approach, we calculated the corresponding 95% CIs for these metrics. Therefore, we used the HSROC model to jointly estimate the classical metrics of sensitivity, specificity, diagnostic ratio, positive likelihood ratio, and negative likelihood ratio for the combined quantities of diagnostic tests, as well as the corresponding 95% CIs. HSROC curves were plotted over the observed range of FP rates for each method, showing the diagnostic effect at different thresholds of positivity [27].

LR compared the probability that test results are similar in people with and those without the target disease. For example, LR+ compared the probability of testing positive for malnutrition among malnourished people with the probability of testing positive for malnutrition among non-malnourished people, i.e. (LR+ = TP rate/FP rate). The negative likelihood ratio (LR-) is (LR- = FN rate/TN rate) [28,29]. In our report, an important feature of LR is the use of Bayes' theorem to estimate the post-test probability. First, we evaluated the pre-test probability as 40% and then calculated the pre-test odds ratio by dividing the pre-test probability by (1—pre-test probability). Next, we obtained the post-test odds ratio by multiplying the pre-test odds ratio by the LR, thus calculating the post-test probability as the post-test odds ratio divided by (1 + post-test odds ratio), and we plotted these results in a Fagan plot [30,31].

The statistical analyses were conducted using Stata software (version 17.0) and Review Manager 5.3(Stata commands for modeling and analysis are available in S4 File). The meta-analysis was registered on the International Prospective Register of Systematic Reviews (PROSPERO) website with the registration number CRD42024519765. The Preferred Reporting Items for Systematic reviews and Meta-Analyses (PRISMA) Statement, which serves as a comprehensive guideline for performing systematic reviews and meta-analyses, was utilized as a point of reference.

## Results

### Characterization and quality assessment of included studies

This meta-analysis included a total of 33 articles that examined six tools for nutritional screening according to the specified criteria (Fig 1). Among them, 8 [32–39] articles on MUST included 10 datasets, 3 [35,36,38] on MST had 4 datasets, 30 [32,33,35,37–39,40–63] on NRS-2002 included 32 datasets, 8 on MNA-SF [33,35,37–39,45,54,63] included 8 datasets, 6 [34–

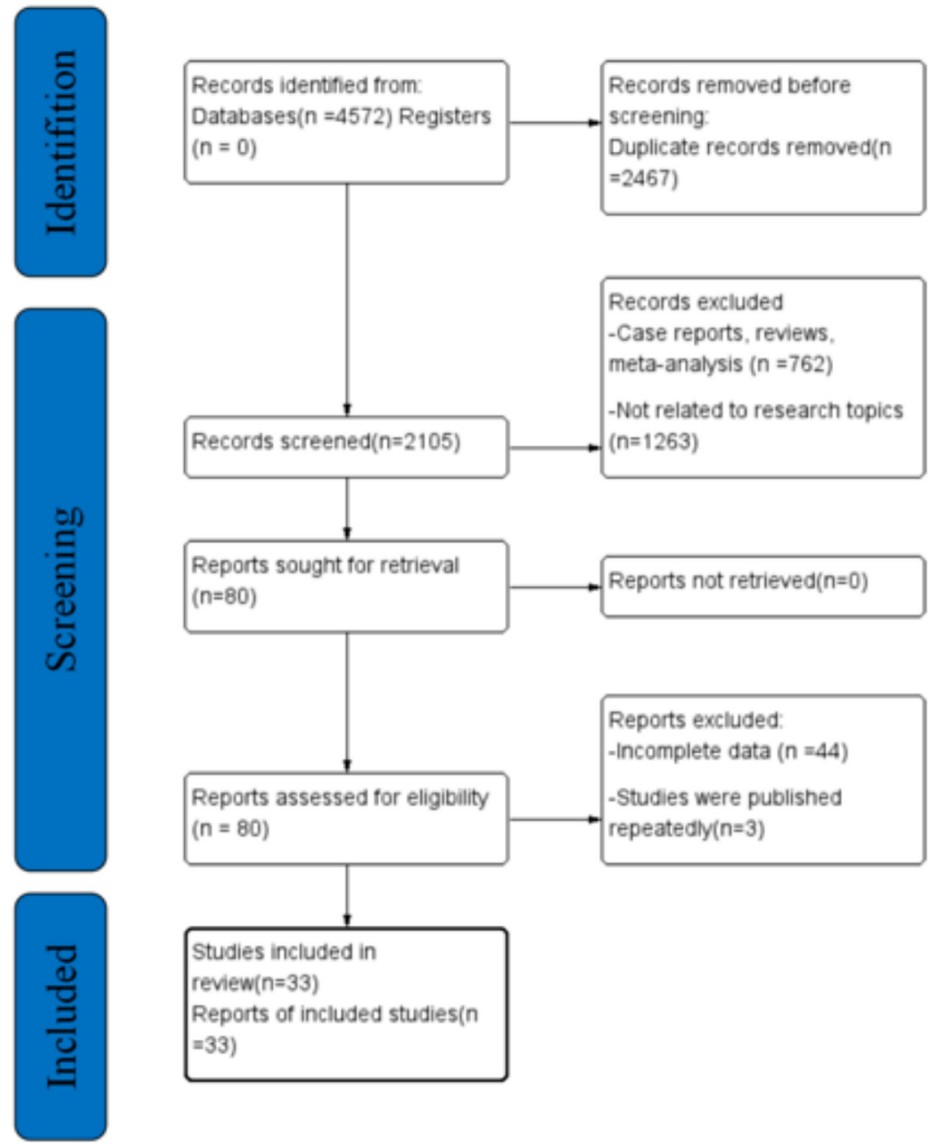

**Fig 1. Flow chart of study selection for inclusion in the meta-analysis.**

36,53,54,64] on NRI had 8 datasets, and 12 [34,38,47,48,55–62] on PG-SGA had 14 datasets (The study screening process and exclusion reasons are detailed in S1 File). Overall, 33 original articles were included in this study, of which 28 were from mainland China, and the remaining five articles were from Iran, Taiwan, Korea, Brazil, and Turkey. Sample sizes ranged from 45 to 1358, totaling 2223 MUST, 578 MST, 9169 NRS-2002, 2083 MNA-SF, 716 NRI, and 5022 PG-SGA patients. Table 1 shows the detailed characteristics of the included studies(Extracted data for meta-analyses are included in S2 File).

**Table 1. Characteristics of included studies.**

| Author | Year | Country | Sample size | Male-female ratio | Age | Diagnosis | Reference standard | Index test | TP | FP | FN | TN |
|---|---|---|---|---|---|---|---|---|---|---|---|---|
| Zibing Wang | 2021 | China | 63 | 63/0 | 62.4 ± 8.2 | Preoperative patient with oesophageal cancer | PG-SGA (A+B/C) | NRS-2002≥3 | 18 | 17 | 2 | 26 |
| Ting Guo | 2015 | China | 100 | 57/43 | 59.7 ± 11.9 | Patients with colorectal cancer | PG-SGA (A+B/C) | NRS-2002≥3 | 14 | 29 | 3 | 54 |
| Changli Wang | 2021 | China | 248 | 158/90 | 56.4 ± 11.9 | Patients with gastric carcinoma | PG-SGA (A+B/C) | NRS-2002≥3 | 105 | 48 | 9 | 86 |
| Daolai Huang | 2018 | China | 181 | 115/66 | 55.29 ± 12.18 | Patients with gastric carcinoma | PG-SGA (A+B/C) | NRS-2002≥3 | 55 | 25 | 21 | 80 |
| Yuqiang Liu | 2017 | China | 99 | 52/47 | 62.25 ± 9.63 | Patients with digestive system tumors undergoing chemotherapy | PG-SGA (A/B+C) | NRS-2002≥3 | 35 | 2 | 15 | 47 |
| Weiping Guo | 2010 | China | 314 | 201/113 | 60 (24–94) | Preoperative patient with gastric cancer | SGA (A/B+C) | MUST≥1 | 125 | 42 | 46 | 101 |
| | | | | | | | | NRS-2002≥3 | 148 | 37 | 23 | 106 |
| Ping Liu | 2013 | China | 80 | 66/14 | 55.5 ± 5.70 | Patients with primary liver cancer | SGA (A/B+C) | MNA<24 | 18 | 18 | 0 | 44 |
| | | | | | | | | NRS-2002≥3 | 17 | 10 | 1 | 52 |
| Yu Zhou | 2017 | China | 196 | 138/58 | 59.32 ± 11.86 | Inpatient with chemotherapy for gastric cancer | PG-SGA (A/B+C) | NRS-2002≥3 | 71 | 3 | 87 | 35 |
| Xiaojing Li | 2018 | China | 103 | 56/47 | 54.480 ± 7.953 | Gastric cancer before postoperative chemotherapy | PG-SGA (≥4) | abPG-SGA >2 | 51 | 9 | 7 | 36 |
| | | | | | | | | NRS-2002≥3 | 35 | 21 | 23 | 24 |
| | | | | | | Middle stage of postoperative chemotherapy for gastric cancer | | abPG-SGA>2 | 60 | 6 | 8 | 29 |
| | | | | | | | | NRS-2002≥3 | 45 | 13 | 23 | 22 |
| | | | | | | Gastric cancer in the late stage of postoperative chemotherapy | | abPG-SGA>2 | 72 | 2 | 4 | 25 |
| | | | | | | | | NRS-2002≥3 | 57 | 8 | 19 | 19 |
| Wan Zhou | 2015 | China | 150 | U | 58.47 ± 9.57 | Gastrointestinal tumor chemotherapy patients | PG-SGA (≥4) | abPG-SGA>2 | 75 | 1 | 6 | 68 |
| | | | | | | | | NRS-2002≥3 | 69 | 15 | 12 | 54 |
| Hai Liang | 2020 | China | 392 | 248/144 | 62.55 ± 11.35 | Inpatients with colorectal cancer | PG-SGA (≥4) | NRS-2002≥3 | 196 | 12 | 96 | 88 |
| Yage Zhu | 2021 | China | 115 | 81/34 | 51 ± 13 | Inpatient with primary liver cancer | SGA (A/B+C) | NRS-2002≥3 | 65 | 2 | 18 | 30 |
| Guibin Li | 2019 | China | 187 | 117/70 | 56.96 ± 13.40 | patients with gastric carcinoma | PG-SGA (≥4) | NRS-2002≥3 | 82 | 4 | 52 | 49 |
| Juntao Chi | 2017 | China | 280 | 166/114 | 62.9 ± 11.9 | Patients undergoing surgery for gastrointestinal tumors | SGA (A/B+C) | NRS-2002≥3 | 89 | 60 | 6 | 125 |
| Shanjun Tan | 2022 | China | 706 | U | U | Patients with stomach/ colorectal cancer | SGA (A/B+C) | NRS-2002≥3 | 108 | 13 | 177 | 408 |
| | | | | | | | | MNA-SF≤11 | 205 | 72 | 80 | 349 |
| | | | | | | | | MUST≥1 | 182 | 46 | 103 | 375 |
| Elnaz Faramarzi | 2012 | Iran | 52 | 40/20 | 54.1 ± 16.8 | Patients with colorectal cancer | PG-SGA (A/B+C) | NRI<100 | 18 | 10 | 9 | 15 |
| Mei-Yu Tu | 2012 | Taiwan | 45 | 25/20 | 62.1 ± 11.5 | Patients with colorectal cancer | SGA (A/B+C) | NRI<100 | 15 | 9 | 1 | 20 |
| | | | | | | | | MUST≥1 | 15 | 5 | 1 | 24 |
| | | | | | | | PA≤20 mg/dL | MUST≥1 | 12 | 8 | 9 | 16 |
| | | | | | | | | NRI<100 | 17 | 7 | 4 | 17 |
| | | | | | | | | SGA (A/B+C) | 13 | 3 | 8 | 21 |

*(Continued)*

**Table 1.** (Continued)

| Author | Year | Country | Sample size | Male-female ratio | Age | Diagnosis | Reference standard | Index test | TP | FP | FN | TN |
|---|---|---|---|---|---|---|---|---|---|---|---|---|
| **Seung Wan Ryu** | 2010 | Korea | 80 | 43/37 | 58.5 ± 11.9 | Patients with gastric carcinoma | SGA (A/B+C) | NRS-2002≥3 | 24 | 11 | 1 | 44 |
| | | | | | | | | NRI<100 | 10 | 15 | 15 | 40 |
| **Bingxin Xie** | 2022 | China | 301 | 178/123 | 62.78 ± 10.56 | Patients with colorectal cancer undergoing surgery | SGA (A/B+C) | NRS-2002≥3 | 99 | 36 | 32 | 134 |
| | | | | | | | | MNA-SF≤11 | 97 | 29 | 34 | 141 |
| | | | | | | | | MUST≥1 | 96 | 41 | 35 | 129 |
| | | | | | | | | MST≥2 | 108 | 45 | 23 | 125 |
| | | | | | | | | NRI<100 | 79 | 58 | 52 | 112 |
| **Mariana Abe Vicente** | 2013 | Brazil | 75 | 36/39 | 60.2+12.2 | Patients with gastric/colorectal cancer before surgery | PG-SGA (A/B+C) | NRI<100 | 34 | 9 | 16 | 16 |
| | | | | | | | | MST≥2 | 26 | 4 | 24 | 21 |
| | | | | | | | | MUST≥1 | 36 | 13 | 14 | 12 |
| | | | 62 | 28/34 | 61.3+11.6 | Patients with gastric/colorectal cancer after surgery | PG-SGA (A/B+C) | NRI<100 | 7 | 8 | 6 | 41 |
| | | | | | | | | MST≥2 | 8 | 4 | 5 | 45 |
| | | | | | | | | MUST≥1 | 11 | 13 | 2 | 36 |
| **Taobo Jin** | 2010 | China | 56 | 35/21 | 58.1 ± 10.1 | Preoperative patient with gastric cancer | ALB<35 g/L | NRS-2002≥3 | 13 | 6 | 3 | 34 |
| | | | | | | | | NRI<100 | 16 | 12 | 0 | 28 |
| | | | | | | | | MNA-SF≤11 | 12 | 26 | 4 | 14 |
| **Yingying Shi** | 2019 | China | 168 | 130/38 | 61 (24–84) | Patients with gastric carcinoma | ALB≤30 g/L | NNRS-2002≥3 | 5 | 77 | 0 | 86 |
| | | | | | | | | PG-SGA≥4 | 5 | 124 | 0 | 39 |
| **Hong Ji** | 2023 | China | 76 | 52/24 | 54.72 ± 8.13 | Patients with liver cancer | the ESPEN diagnostic criteria | NRS-2002≥3 | 51 | 2 | 18 | 5 |
| | | | | | | | | PG-SGA (A/B+C) | 64 | 1 | 5 | 6 |
| **Li Lin** | 2018 | China | 680 | 384/296 | 57.2 ± 9.0 | Patients with digestive system tumors | ALB≤30 g/L | NRS-2002≥3 | 192 | 62 | 44 | 376 |
| | | | | | | | | PG-SGA≥4 | 205 | 41 | 37 | 397 |
| **Xi Qiao** | 2015 | China | 457 | 363/105 | 59.04 ± 10.1 | Patients with advanced gastric cancer | BMI <18.5 kg/m$^2$/ALB<35 g/L | NRS-2002≥3 | 104 | 137 | 10 | 216 |
| | | | | | | | | PG-SGA (A/B+C) | 108 | 248 | 6 | 105 |
| **Bingxin Xie** | 2022 | China | 280 | 164/116 | U | Patients with colorectal cancer undergoing surgery | BMI<18.5 kg/m$^2$/ALB<35 g/L | NRS-2002≥3 | 35 | 81 | 12 | 152 |
| | | | | | | | | MUST≥1 | 30 | 84 | 17 | 149 |
| | | | | | | | | MNA-SF≤11 | 34 | 95 | 13 | 138 |
| **Xite Zheng** | 2024 | China | 1308 | 933/375 | 60 (52–68) | Patients with gastric carcinoma | The fitting Bayesian LCM analysis | NRS-2002≥3 | 646 | 38 | 348 | 276 |
| | | | | | | | | PG-SGA (A/B+C) | 954 | 41 | 40 | 273 |
| **Shengqiang Tan** | 2024 | China | 207 | 170/37 | 56.6 ± 11.3 | Patients with liver cancer | The GLIM diagnostic criteria | NRS-2002≥3 | 57 | 6 | 32 | 112 |
| | | | | | | | | PG-SGA≥4 | 84 | 29 | 5 | 89 |
| **Xiaoli Ruan** | 2022 | China | 1358 | 810/548 | 60 (52–67) | Patients with colorectal cancer | The fitting Bayesian LCM analysis | NRS-2002≥3 | 486 | 60 | 274 | 538 |

*(Continued)*

**Table 1.** (Continued)

| Author | Year | Country | Sample size | Male-female ratio | Age | Diagnosis | Reference standard | Index test | TP | FP | FN | TN |
|---|---|---|---|---|---|---|---|---|---|---|---|---|
| | | | | | | | | PG-SGA (A/B+C) | 730 | 108 | 30 | 490 |
| Reyyan Yildirim | 2020 | Turkey | 140 | 100/40 | 64.2 ± 11.8 | Patients with gastric undergoing surgery | The ESPEN diagnostic criteria | NRS-2002≥3 | 23 | 53 | 6 | 58 |
| | | | | | | | | MUST≥2 | 25 | 34 | 4 | 77 |
| | | | | | | | | SGA (A/B+C) | 19 | 40 | 10 | 71 |
| | | | | | | | | MNA-SF≤11 | 25 | 18 | 4 | 93 |
| | | | | | | | | MST≥2 | 25 | 67 | 4 | 44 |
| Dong Yang | 2020 | China | 114 | 61/53 | 57.1 ± 13.2 | Gastric cancer with pyloric obstruction | BMI≤18.5 kg/m²/ALB≤30 g/L | NRS-2002≥3 | 62 | 8 | 18 | 26 |
| | | | | | | | | PG-SGA≥4 | 71 | 5 | 9 | 29 |
| Xiao-Jun Ye | 2018 | China | 255 | 160/95 | 76.5 ± 408 | Patients with gastrointestinal tumors | the ESPEN diagnostic criteria | NRS-2002≥3 | 47 | 86 | 4 | 118 |
| | | | | | | | | MUST≥1 | 48 | 48 | 3 | 156 |
| | | | | | | | | MNA-SF≤11 | 48 | 74 | 3 | 130 |
| Qianqian Zhang | 2021 | China | 265 | 193/72 | 70 (66–74) | Inpatient with gastrointestinal cancer | ALB≤40 g/L | NRS-2002≥3 | 84 | 98 | 26 | 57 |
| | | | | | | | | MNA-SF≤11 | 80 | 97 | 30 | 58 |

MUST, Malnutrition Universal Screening Tool; MST, Malnutrition Screening Tool; NRS-2002, Nutritional Risk Screening 2002; MNA-SF, Mini Nutritional Assessment-Short Form; NRI, Nutritional Risk Index; PG-SGA, Patient-Generated Subjective Global Assessment; ESPEN, European Society for Clinical Nutrition and Metabolism; GLIM, Global Leadership Initiative on Malnutrition; BMI, body mass index; abPG-SGA, Abridged Scored Patient-Generated Subjective Global Assessment; ALB, albumin; SGA, Subjective Global Assessment; TP, true positive; FP, false positive; FN, false negative; TN, true negative

The literature collected in this study was deemed appropriate based on a methodological quality assessment utilizing the QUADAS checklist (Fig 2) (Risk of bias and quality assessment results are provided in S3 File).

## Meta-analysis results

Table 2 displays the combined calculations of sensitivity, specificity, and LR acquired by the HSROC model. The main metrics used to assess the efficacy of the diagnostic tests were sensitivity and specificity. The MUST showed a sensitivity of 0.765 (95% CI: 0.675–0.835) and a specificity of 0.736 (95% CI: 0.665–0.797). The MST had a sensitivity of 0.739 (95% CI: 0.539–0.873) and a specificity of 0.773 (95% CI: 0.524–0.913). NRS-2002 displayed a sensitivity of 0.778 (95% CI: 0.725–0.823) and a specificity of 0.784 (95% CI: 0.721–0.835). MNA-SF demonstrated a sensitivity of 0.806 (95% CI: 0.712–0.875) and a specificity of 0.669 (95% CI: 0.528–0.784). The NRI had a sensitivity of 0.725 (95% CI: 0.553–0.849) and a specificity of 0.692 (95% CI: 0.636–0.744), while the PG-SGA exhibited a sensitivity of 0.911 (95% CI: 0.866–0.942) and a specificity of 0.805 (95% CI: 0.674–0.891). Out of all assessment instruments, the PG-SGA showed the greatest sensitivity and specificity, while the NRI and MNA-SF had the lowest sensitivity and specificity, respectively. The HSROC curves provide a comprehensive representation of the anticipated diagnostic accuracy regarding sensitivity and specificity, as depicted in Fig 3. Sensitivity and specificity for MUST, MST, NRS-2002, MNA-SF, NRI, and PG-SGA are represented in the forest plots across all included studies, as shown in Fig 4.

The LR+ and LR- with their corresponding CI for various screening tools are as follows: MUST: LR+ of 2.901 (95% CI: 2.207–3.812) and LR- of 0.320 (95% CI: 0.224–0.456); MST:

(a)

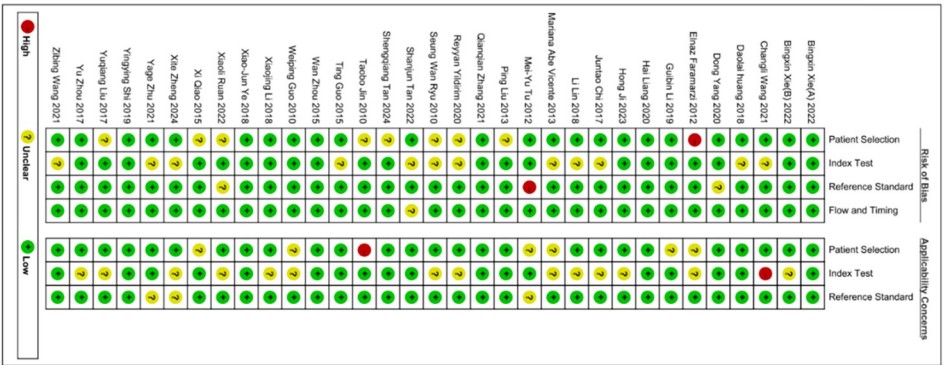

(b)

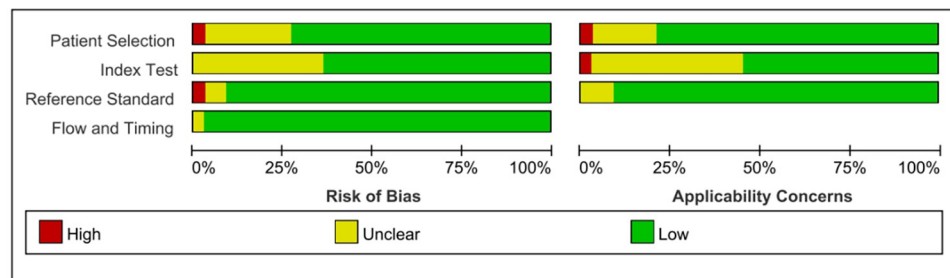

**Fig 2. Risk assessment of included studies using QUADAS.** (a) Risk of bias and applicability concerns summary: review authors' judgments about each domain for each included study. (b) Risk of bias and applicability concerns graph: review authors' judgments about each domain presented as percentages across included studies.

**Table 2. Result of meta-analysis and Bayes analysis.**

|  | MUST | MST | NRS-2002 | MNA-SF | NRI | PG-SGA |
|---|---|---|---|---|---|---|
| **Sensitivity (95% CI)** | 0.765 (95% CI: 0.675–0.835) | 0.739 (95% CI: 0.539–0.873) | 0.778 (95% CI: 0.725–0.823) | 0.806 (95% CI: 0.712–0.875) | 0.725 (95% CI: 0.553–0.849) | 0.911 (95% CI: 0.866–0.942) |
| **Specificity (95% CI)** | 0.736 (95% CI: 0.665–0.797) | 0.773 (95% CI: 0.524–0.913) | 0.784 (95% CI: 0.721–0.835) | 0.669 (95% CI: 0.528–0.784) | 0.692 (95% CI: 0.636–0.744) | 0.805 (95% CI: 0.674–0.891) |
| **DOR (95% CI)** | 9.074 (95% CI: 5.150–15.987) | 9.640 (95% CI: 4.626–20.088) | 12.686 (95% CI: 9.387–17.145) | 8.399 (95% CI: 3.717–18.978) | 5.948 (95% CI: 2.724–12.988) | 41.987 (95% CI: 18.870–93.424) |
| **LR+(95% CI)** | 2.901 (95% CI: 2.207–3.812) | 3.251 (95% CI: 1.596–6.625) | 3.596 (95% CI: 2.840–4.555) | 2.433 (95% CI: 1.608–3.680) | 2.360 (95% CI: 1.815–3.068) | 4.66 (95% CI: 2.680–8.108) |
| **LR-(95% CI)** | 0.320 (95% CI: 0.224–0.456) | 0.337 (95% CI: 0.206–0.551) | 0.283 (95% CI: 0.234–0.343) | 0.290 (95% CI: 0.180–0.466) | 0.400 (95% CI: 0.229–0.687) | 0.111 (95% CI: 0.072–0.171) |
| **1/LR-(95% CI)** | 3.128 (95% CI: 2.194–4.460) | 2.965 (95% CI: 1.815–4.844) | 3.527 (95% CI: 2.916–4.267) | 3.452 (95% CI: 2.145–5.556) | 2.521 (95% CI: 1.456–4.364) | 9.007 (95% CI: 5.861–13.842) |

CI, confidence interval; LR+ positive likelihood ratios; LR-, negative likelihood ratios; MUST, Malnutrition Universal Screening Tool; MST, Malnutrition Screening Tool; NRS-2002, Nutritional Risk Screening 2002; MNA-SF, Mini Nutritional Assessment-Short Form; NRI, Nutritional Risk Index; PG-SGA, Patient-Generated Subjective Global Assessment.

(a)

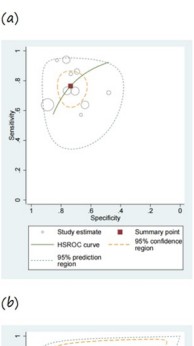

(b)

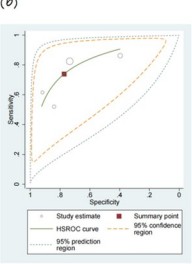

(c)

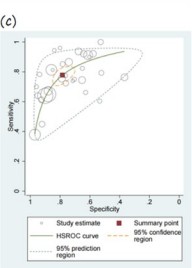

(d)

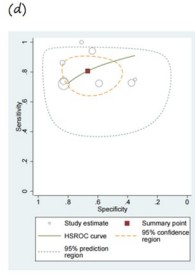

(e)

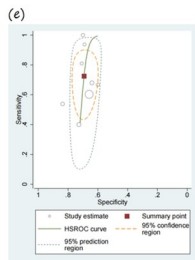

(f)

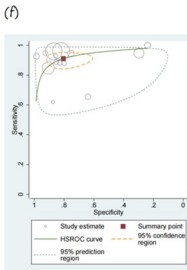

**Fig 3. Hierarchical summary receiver operating characteristic (HSROC) curves.** (a) MUST. (b) MST. (c) NRS-2002. (d) MNA-SF. (e) NRI. (f) PG-SGA. MUST, Malnutrition Universal Screening Tool; MST, Malnutrition Screening Tool; NRS-2002, Nutritional Risk Screening 2002; MNA-SF, Mini Nutritional Assessment-Short Form; NRI, Nutritional Risk Index; PG-SGA, Patient-Generated Subjective Global Assessment.

LR+ of 3.251 (95% CI: 1.596–6.625) and LR- of 0.337 (95% CI: 0.206–0.551); NRS-2002: LR + of 3.596 (95% CI: 2.840–4.555) and LR- of 0.283 (95% CI: 0.234–0.343); MNA-SF: LR+ of 2.433 (95% CI: 1.608–3.680) and LR- of 0.290 (95% CI: 0.180–0.466); NRI: LR+ of 2.360 (95% CI: 1.815–3.068) and LR- of 0.400 (95% CI: 0.229–0.687); PG-SGA: LR+ of 4.66 (95% CI: 2.680–8.108) and LR- of 0.111 (95% CI: 0.072–0.171). Among the evaluation tools, the PG-SGA exhibited the highest LR+ and the lowest LR-, followed by the NRS-2002. On the other hand, the NRI had the lowest LR+ and the highest LR-.

## Bayesian analysis

Based on the analysis of the Fagan plots in Fig 5, we evaluated the initial 40% pretest probability and its resulting post-test probability as follows: 66% post-test probability and 18% negative post-test probability for MUST; 68% post-test probability and 18% negative post-test probability for MST; 71% post-test probability and 16% negative post-test probability for NRS-2002; 62% post-test probability and 16% negative post-test probability for MNA-SF; 61% post-test probability after a positive test result and 21% in a negative scenario for NRI; and 76% post-test probability after a positive test result and 7% in a negative scenario for PG-SGA. This indicates that PG-SGA is the most detailed approach, with a 76% probability after a positive test. In addition, the likelihood of malnutrition decreased in patients who had negative PG-SGA results.

## Publication bias

Deek's method was used to evaluate publication bias in each study technique. Findings indicated that MUST (P = 0.79), MST (P = 0.59), NRS-2002 (P = 0.58), MNA-SF (P = 0.91), NRI (P = 0.14), and PG-SGA (P = 0.18) exhibited no significant publication bias (P>0.05). This shows that the outcomes of these investigations are not impacted by selective reporting or publication bias (Fig 6).

## Discussion

Malignant tumors of the digestive system can severely affect food intake, digestion and nutrient absorption, resulting in people diagnosed with these tumors being more likely to be deficient in nutrients and more susceptible to malnutrition [65]. Scientific and rational nutritional therapy can improve the nutritional status of patients and increase their tolerance to treatment, and accurate nutritional assessment is the first step in nutritional therapy. Various tools are used in clinical practice to assess nutritional status, each of which is tailored to a specific population and has a different focus and specificity. Accurate assessment of nutritional status is essential to guide interventions and improve the prognosis of patients with gastrointestinal tumors. In this study, we assessed the reliability of nutritional screening techniques in patients with gastrointestinal malignancies using a hierarchical Bayesian latent class meta-analysis. Our data suggest that the PG-SGA tool has the highest sensitivity and specificity and is the most accurate tool for diagnosing malnutrition in this group of patients. The findings further confirm previous studies that have highlighted the accuracy of the PG-SGA tool in identifying malnutrition. [66,67]. The PG-SGA is a nutritional risk assessment tool designed by Autry specifically for cancer patients and is an improvement on the SGA. It provides a comprehensive

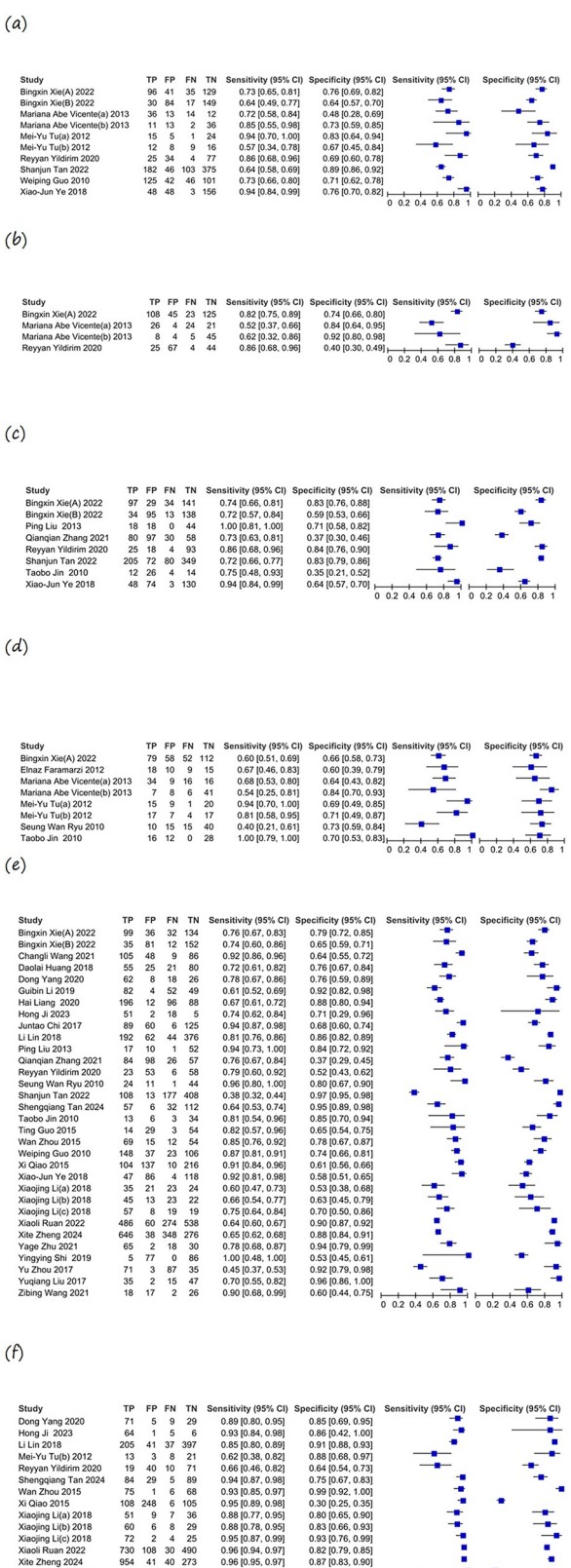

**Fig 4. Forest plot for sensitivity and specificity.** (a) MUST. (b) MST. (c) MNA-SF. (d) NRI. (e) NRS-2002. (f) PG-SGA. MUST, Malnutrition Universal Screening Tool; MST, Malnutrition Screening Tool; NRS-2002, Nutritional Risk Screening 2002; MNA-SF, Mini Nutritional Assessment-Short Form; NRI, Nutritional Risk Index; PG-SGA, Patient-Generated Subjective Global Assessment.



**Fig 5. Fagan plot analysis.** (a) MUST. (b) MUT. (c) NRS-2002. (d) MNA-SF. (e) NRI. (f) PG-SGA. MUST, Malnutrition Universal Screening Tool; MST, Malnutrition Screening Tool; NRS-2002, Nutritional Risk Screening 2002; MNA-SF, Mini Nutritional Assessment-Short Form; NRI, Nutritional Risk Index; PG-SGA, Patient-Generated Subjective Global Assessment.

assessment of a patient's personal satisfaction and nutritional status [68]. The comprehensive assessment approach of the PG-SGA takes into account a number of aspects, including weight fluctuations, dietary intake, symptoms, functional status, and metabolic needs, in order to measure a patient's nutritional status. It is a reliable nutritional assessment tool that allows for precise interventions and improves patient recognition of the risk of malnutrition. The Dietitians Association of Australia specifically recommends the use of the PG-SGA Nutritional Risk Screening Scale to assess the nutritional needs of cancer patients undergoing radiotherapy [69]. The American Dietetic Association has designated the PG-SGA Nutritional Risk Screening Scale as the preferred tool for nutritional risk screening in patients with cancer [70]. The Chinese Society of Clinical Oncology Expert Consensus on Nutrition Therapy recognizes the PG-SGA Nutritional Risk Screening Scale as a widely used tool for screening nutritional risk in cancer patients [71]. The revised Abridged Scored Patient-Generated Subjective Global Assessment (abPG-SGA) [72] is a condensed version of the PG-SGA, designed to reduce the duration of assessment for healthcare practitioners and enhance the efficiency of nutrition screening. Multiple studies have shown that the abPG-SGA has high sensitivity and specificity, demonstrates a strong correlation with the PG-SGA, and is easy to administer. Additionally, it is particularly suitable for assessing the nutritional status of patients undergoing chemotherapy for gastrointestinal tumors [48,49]. Contrastingly, the NRI tool is less sensitive. This may be because it relies primarily on biomarkers (e.g., albumin and prealbumin) to assess nutritional status, which may be affected by a variety of factors such as inflammation, liver function, and edema [73]. Therefore, NRI may not fully reflect the true nutritional status of patients.

Due to the lack of a recognized gold standard, different studies have used different reference standards. This variation has led to different results, thus overestimating or underestimating the accuracy of the tests and contributing to the heterogeneity of some of the results. In our study, there were 5 reference standards for MUST, 3 for MST, 12 for NRS-2002, 5 for MNA-SF, 4 for NRI, and 8 for PG-SGA. Some criteria were considered inappropriate, including serum hepatic protein levels, such as albumin, which, while reflecting disease severity, are not an accurate indicator of nutritional status [74,75]. In addition, we examined two articles using Bayesian latency modeling to estimate the occurrence of malnutrition in patients with stomach and intestinal malignancies and the accuracy of nutritional assessment methods [59,60]. Bayesian analysis provides a powerful tool that can help clinicians estimate based on screening results. However, whether it can be accurately and effectively used as a criterion for judgment remains to be confirmed by further studies. In this study, we used an appropriate and effective method to model the variability of the reference test and adjust for its limitations. We used the HSROC model to address the problem of reference standard variation across studies. This model can better account for LR and post-test probabilities, thereby improving our understanding of the accuracy of nutritional screening tools.

Although this study provides some valuable insights, it has some limitations. First, various factors, including patient characteristics and the scope of the included studies, may influence the variability of the study results. Particularly, this study focused on literature in mainland China, which may limit the generalizability of the findings. Additionally, it should be specifically emphasized that the results of various studies may be affected by differences in study design, sample characteristics, and study quality, and even though the HSROC modeling approach was used to address heterogeneity, the effect of heterogeneity on the results could

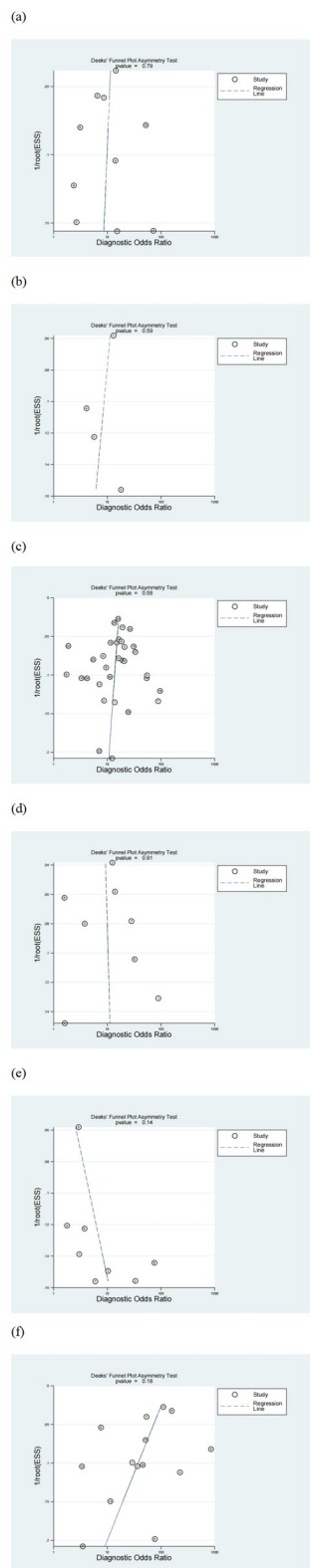

**Fig 6. Deeks' funnel plot.** (a) MUST. (b) MST. (c) NRS-2002. (d) MNA-SF. (e) NRI. (f) PG-SGA. MUST, Malnutrition Universal Screening Tool; MST, Malnutrition Screening Tool; NRS-2002, Nutritional Risk Screening 2002; MNA-SF, Mini Nutritional Assessment-Short Form; NRI, Nutritional Risk Index; PG-SGA, Patient-Generated Subjective Global Assessment.

not be completely eliminated. Therefore, the dependence on a priori probability and the complexity of the model should be further investigated and improved [76]. Lian et al [77] extended the Bayesian HSROC model to propose an NMA method for diagnostic accuracy tests that can handle missing data. The method can incorporate data from multiple design types, considering the presence or absence of a gold standard, heterogeneity between studies, and correlations between diagnostic test accuracy metrics. Additionally, the authors found through simulation studies that this method makes better use of the data and is also more effective than the HSROC Meta-regression [78] method. However, the model is based on the consistency assumption, which currently cannot resolve inconsistencies between direct evidence, and they have not made this publicly available. Therefore, when interpreting the findings, the possible impact of heterogeneity between studies on the results to ensure the reliability and generalizability of the conclusions should be carefully considered. To further improve the validation and refinement of nutritional screening tools, future studies should include a wider geographic area and explore harmonized assessment criteria to better understand the applicability and validity of nutritional screening tools and provide more practical guidance for clinical practice.

## Conclusion

This meta-analysis evaluated the accuracy of various nutritional screening tools for patients with gastrointestinal tumors, finding that the Patient-Generated Subjective Global Assessment (PG-SGA) exhibits exceptionally high sensitivity and specificity. As a well-established and reliable method, PG-SGA provides a thorough assessment of malnutrition through comprehensive patient evaluation. While other nutritional screening tools have value in specific clinical settings, their effectiveness may vary depending on patient characteristics and clinical requirements. Thus, when selecting an appropriate nutritional screening tool, clinicians should consider these factors along with the specific strengths and limitations of each instrument.

Given the study's focus on a predominantly Chinese population, future research should aim to encompass a broader geographic scope. This would improve the generalizability of the findings and account for cultural and dietary variations that may influence nutritional risk. Additionally, the development of standardized assessment criteria across studies will help reduce variability and bias, ultimately enhancing the effectiveness of nutritional screening tools in clinical practice.

## Supporting information

**S1 File. Study screening process and exclusion reasons.** This file provides a detailed table documenting the inclusion and exclusion of studies during the systematic review, along with the reasons for exclusion.
(DOC)

**S2 File. Data extracted from included studies.** This file contains the data extracted from the included studies, including study characteristics, diagnostic performance metrics, and other relevant details used in the meta-analyses.
(DOC)

**S3 File. Risk of bias and quality assessment (cochrane risk of bias tool).** This file presents the results of the risk of bias and quality assessments for the included studies, conducted using standardized evaluation tools.
(DOC)

**S4 File. Stata commands used in the analysis.** This file provides the Stata commands and scripts used for the hierarchical Bayesian modeling and statistical analyses performed in this

study.
(DOC)

## Author Contributions

**Conceptualization:** Xiuqin Yang.

**Data curation:** Menghao Yang, Na Xiao.

**Project administration:** Yuexiu Wen.

**Software:** Menghao Yang, Na Xiao.

**Supervision:** Xiuqin Yang.

**Validation:** Le Tang, Yang Zhang.

**Visualization:** Menghao Yang, Na Xiao.

**Writing – original draft:** Menghao Yang, Na Xiao.

**Writing – review & editing:** Menghao Yang, Na Xiao.

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
