## [Decision Letter · Decision Letter 0]

22 Sep 2024

PONE-D-24-31460Evaluating the accuracy of a nutritional screening tool for patients with digestive system tumors: A hierarchical Bayesian latent class meta-analysisPLOS ONE

Dear Dr. Yang,

Thank you for submitting your manuscript to PLOS ONE. After careful consideration, we feel that it has merit but does not fully meet PLOS ONE’s publication criteria as it currently stands. Therefore, we invite you to submit a revised version of the manuscript that addresses the points raised during the review process:

**Please, re-adjust your Abstract and Conclusion in consideration of the comments done by the reviewer.** 

We look forward to receiving your revised manuscript.

Kind regards,

Domenico Fuoco

Academic Editor

PLOS ONE

Journal Requirements:

2. Thank you for stating the following financial disclosure: This study received funding from the Science and Technology Fund Project of the Health and Wellness Commission of Guizhou Province, China, under Project No. [gzwkj2024-266].

4. As required by our policy on Data Availability, please ensure your manuscript or supplementary information includes the following: 

Reviewers' comments:

Reviewer's Responses to Questions

**Comments to the Author**

1. Is the manuscript technically sound, and do the data support the conclusions?

Reviewer #1: Partly

2. Has the statistical analysis been performed appropriately and rigorously? 

Reviewer #1: I Don't Know

3. Have the authors made all data underlying the findings in their manuscript fully available?

Reviewer #1: Yes

4. Is the manuscript presented in an intelligible fashion and written in standard English?

Reviewer #1: Yes

5. Review Comments to the Author

Reviewer #1: There are some valuable insights in the study, it also raises some "food for thought". As most studies in its initial stages, it has it imitations. The people in the study group are localized in an area of China, hence the results are based on the very localized group, which may or may not be true for global conclusions. As some food habits and use of some chemicals or impact of radiation in particular areas can lead to cancer in a particular area.

The study does have an interesting angle to explore and if done on a larger scale in other parts of the world provide more conclusive results that could be universally acceptable.

6. PLOS authors have the option to publish the peer review history of their article (what does this mean?). If published, this will include your full peer review and any attached files.

Reviewer #1: **Yes: **Ashok Bhaseen, M. Pharm, MMS

---

## [Author Response · Author response to Decision Letter 0]

9 Oct 2024

Reviewer #1:

Comment 1: The study’s population is localized in a specific region of China, which may limit the generalizability of the findings.

Response: We acknowledge this limitation and have revised the Abstract and Conclusion sections accordingly. We now explicitly state that the findings are based on a specific Chinese population, which may affect their broader applicability. We have also suggested that future research should aim to validate the effectiveness of these nutritional screening tools in more diverse populations to ensure generalizability.

Comment 2: The statistical methods, particularly the hierarchical Bayesian analysis, require further clarification.

Response: We have expanded the Methods section to include a detailed explanation of the hierarchical Bayesian latent class meta-analysis. This includes the rationale for selecting the hierarchical summary receiver operating characteristic (HSROC) model, which is particularly appropriate for studies involving multiple diagnostic tools. We have also provided a step-by-step explanation of the statistical process, including the use of Stata commands (metandi and midas) to calculate diagnostic performance metrics. These details are further clarified in Supplementary Material 4 to enhance transparency and reproducibility.

Comment 3: Similar studies should be conducted in different parts of the world to provide more universally applicable conclusions.

Response: We agree with the reviewer and have revised the Conclusion section to emphasize the importance of conducting multi-regional studies. This will ensure that the findings from our study can be validated in a broader range of cultural and healthcare settings.

Comment4: Concerns were raised about the references used and the integrity of the data extraction process.

Response: After carefully reviewing all references, we confirm that none of the cited studies have been retracted or are unpublished. All references were verified using major academic databases, including PubMed, Embase, and the Cochrane Library. Regarding data extraction, this was performed by two independent reviewers, and the process was carefully cross-verified to ensure accuracy. No external or unpublished data were used, and a comprehensive list of included and excluded studies (along with reasons for exclusion) is provided in Supplementary Material 1.

Editor Comment: You requested clarification and confirmation regarding the data availability plan.

Response: We have revised the Data Availability Statement as requested. The dataset has been made freely available via the Dryad repository (DOI: doi:10.5061/dryad.4mw6m90m8), ensuring compliance with PLOS ONE’s policies. The dataset includes all extracted data points, such as true positives, false positives, false negatives, and true negatives, along with the Stata commands used in the statistical analysis. The data have been anonymized to maintain ethical compliance.

---

## [Editor Report · Decision Letter 1]

6 Dec 2024

Evaluating the accuracy of a nutritional screening tool for patients with digestive system tumors: A hierarchical Bayesian latent class meta-analysis

PONE-D-24-31460R1

Dear Dr. Yang,

We’re pleased to inform you that your manuscript has been judged scientifically suitable for publication and will be formally accepted for publication once it meets all outstanding technical requirements.

Kind regards,

Sudhakar Jinka, Ph.D.

Guest Editor

PLOS ONE
---

## [Editor Report · Acceptance letter]

11 Dec 2024

PONE-D-24-31460R1 

PLOS ONE

Dear Dr. Yang, 

I'm pleased to inform you that your manuscript has been deemed suitable for publication in PLOS ONE. Congratulations! Your manuscript is now being handed over to our production team.

Kind regards, 

on behalf of

Dr. Sudhakar Jinka 

Guest Editor

PLOS ONE